# PROBABILISTIC MIXTURE-OF-EXPERTS FOR EFFICIENT DEEP REINFORCEMENT LEARNING

## ABSTRACT

Deep reinforcement learning (DRL) has successfully solved various problems recently, typically with a unimodal policy representation. However, grasping the decomposable and hierarchical structures within a complex task can be essential for further improving its learning efficiency and performance, which may lead to a multimodal policy or a mixture-of-experts (MOE). To our best knowledge, present DRL algorithms for general utility do not deploy MOE methods as policy function approximators due to the lack of differentiability, or without explicit probabilistic representation. In this work, we propose a differentiable probabilistic mixture-of-experts (PMOE) embedded in the end-to-end training scheme for generic off-policy and on-policy algorithms using stochastic policies, e.g., Soft Actor-Critic (SAC) and Proximal Policy Optimisation (PPO). Experimental results testify the advantageous performance of our method over unimodal polices and three different MOE methods, as well as a method of option frameworks, based on two types of DRL algorithms. We also demonstrate the distinguishable primitives learned with PMOE in different environments.

## 1 INTRODUCTION

The mixture-of-experts method (MOE) (Jacobs et al., 1991a) is testified to be capable of improving the generalisation ability of reinforcement learning (RL) agents (Hausknecht & Stone, 2016a; Peng et al., 2016; Neumann et al.). Among these methods, the Gaussian Mixture Models (GMM) are promising to model multimodal policy in RL (Peng et al., 2019; Akrour et al., 2020), in which distinguishable experts or so-called primitives are learned. The distinguishable experts can propose several solutions for a task and have a larger range of exploration space, which can potentially lead to better task performance and sample efficiency compared to its unimodal counterpart (Bishop, 2007). The multimodal policy can be learned by various methods, such as a two-stage training approach (Peng et al., 2019), specific clustering method (Akrour et al., 2020), or especially parameterised actions design (Hausknecht & Stone, 2016b). However, these methods are limited, neither applicable to complicated scenarios such as high-dimensional continuous control tasks nor the training algorithms are too complex to deal with general utility. To the best of our knowledge, the present DRL algorithms for general utility do not deploy MOE to model the multimodal policy mainly due to the lack of differentiability, or without explicit probabilistic representation. Therefore, in the policy gradient-based algorithms (Sutton et al., 1999a), the gradient of the performance concerning the policy parameters is undifferentiated. The undifferentiability problem also remains to learn a deep neural network policy thus making the combinations of MOE and DRL not trivial.

In this paper, we propose a probabilistic framework to tackle the undifferentiated problem by holding the mixture distribution assumption. We will still use the GMM to model the multimodal policies. Once the undifferentiated problem is solved, our training methods can be combined with the policy gradient algorithms by simply setting the number of experts (mixtures) greater than one. Hereafter, the contribution can be summarised as follows:

- We analyse the undifferentiability problem of approximating policy as the GMM in DRL and its associated drawbacks.

- We propose an end-to-end training method to obtain the primitives with probability in a frequentist manner to solve the undifferentiability problem.

- Our experiments show the proposed method can achieve better task performance and sample efficiency by exploring larger behaviours space, especially in complicated continuous control tasks, compared with unimodal RL algorithms and three different MOE methods or option frameworks.

## 2 RELATED WORK

**Hierarchical Policies**    There are two main related hierarchical policy structures. The feudal schema (Dayan & Hinton, 1992) has two types of agents: managers and workers. The managers first make high-level decisions, then the workers make low-level actions according to these high-level decisions. The options framework (Sutton et al., 1999b) has an upper-level agent (policy-over-options), which decides whether the lower level agent (sub-policy) should start or terminate. In the early years, it's the subject of research to discover temporal abstractions autonomously often in discrete actions and the state space (McGovern & Barto, 2001; Menache et al., 2002; Simsek & Barto, 2008; Silver & Ciosek, 2012). Recently, (Mankowitz et al., 2016) proposes a method that assumes the initiation sets and termination functions have particular structures. (Kulkarni et al., 2016) uses internal and extrinsic rewards to learn sub-policies and policy-over-options. (Bacon et al., 2017) trains sub-policies and policy-over-options in end-to-end fusion with a deep termination function. (Vezhnevets et al., 2017) generalises the feudal schema into continuous action space and uses an embedding operation to solve the indifferentiable problem. (Peng et al., 2016) introduces a mixture of actor-critic experts approaches to learn terrain-adaptive dynamic locomotion skills. (Peng et al., 2019) changes the mixture-of-experts distribution addition expression into the multiplication expression.

**Mixture-of-Experts and Ensemble Methods**    To speed up the learning and improve the generalisation ability on different scenarios, Jacobs et al. (1991a) proposed to use several different expert networks instead of a single one. To partition the data space and assign different kernels for different spaces, Lima et al. (2007); Yao et al. (2009) combines MOE with SVM. To break the dependency among training outputs and speed up the convergence, Gaussian process (GP) is generalised similarly to MOE (Tresp, 2000; Yuan & Neubauer, 2008; Luo & Sun, 2017). MOE can be also combined with RL (Doya et al., 2002; Neumann et al.; Peng et al., 2016; Hausknecht & Stone, 2016a; Peng et al., 2019), in which the policies are modelled as probabilistic mixture models and each expert aim to learn distinguishable policies.

**Policy-based RL**    Policy-based RL aims to find the optimal policy to maximise the expected return through gradient updates. Among various algorithms, Actor-critic is often employed (Barto et al., 1983; Sutton & Barto, 1998). Off-policy algorithms (O'Donoghue et al., 2016; Lillicrap et al., 2016; Gu et al., 2017; Tuomas et al., 2018) are more sample efficient than on-policy ones (Peters & Schaal, 2008; Schulman et al., 2017; Mnih et al., 2016; Gruslys et al., 2017). However, the learned policies are still unimodal.

## 3 METHOD

### 3.1 NOTATION

The model-free RL problem can be formulated by Markov Decision Process (MDP), denoted as a tuple $(\mathcal{S}, \mathcal{A}, P, r)$, where $\mathcal{S}$ and $\mathcal{A}$ are continuous state and action space, respectively. The agent observes state $s_t \in \mathcal{S}$ and takes an action $a_t \in \mathcal{A}$ at time step $t$. The environment emits a reward $r : \mathcal{S} \times \mathcal{A} \to [r_{min}, r_{max}]$ and transitions to a new state $s_{t+1}$ according to the transition probabilities $P : \mathcal{S} \times \mathcal{S} \times \mathcal{A} \to [0, \infty)$. In deep reinforcement learning algorithms, we always use the Q-value function $Q(s_t, a_t)$ to describe the expected return after taking an action $a_t$ in the state $s_t$. The Q-value can be iteratively computed by applying the Bellman backup given by:

$$Q(s_t, a_t) \triangleq \mathbb{E}_{s_{t+1} \sim P}\big[r(s_t, a_t) + \gamma \mathbb{E}_{a_{t+1} \sim \pi}[Q(s_{t+1}, a_{t+1})]\big]. \tag{1}$$

Our goal is to maximise the expected return:

$$\pi_{\Theta*}(a_t|s_t) = \underset{\pi_{\Theta}(a_t|s_t)}{\arg\max} \mathbb{E}_{a_t \sim \pi_{\Theta}(a_t|s_t)}[Q(s_t, a_t)], \tag{2}$$

where $\Theta$ denotes the parameters of the policy network $\pi$. With Q-value network (critic) $Q_\phi$ parameterised by $\phi$, Stochastic gradient descent (SGD) based approaches are usually used to update the policy network:

$$\Theta = \Theta + \nabla_\Theta \mathbb{E}_{a \sim \pi_\Theta(a_t|s_t)}[Q_\phi(s_t, a_t)]. \tag{3}$$

## 3.2 PROBABILISTIC MIXTURE-OF-EXPERTS (PMOE)

The proposed PMOE method decomposes a stochastic policy $\pi$ as a mixture of low-level policies while retaining the probabilistic properties of the stochastic policy as a probability distribution, with the following formula:

$$\pi_{\{\theta,\psi\}}(a_t|s_t) = \sum_{i=1}^{K} w_{\theta_i}(s_t)\pi_{\psi_i}(a_t|s_t), \; s.t. \sum_{i=1}^{K} w_{\theta_i} = 1, \; w_{\theta_i} > 0, \tag{4}$$

where each $\pi_{\psi_i}$ denotes the action distribution within each low-level policy, *i.e.* a *primitive*, and $K$ denotes the number of primitives. $w_{\psi_i}$ is the weight that specifies the probability of the activating primitive $\pi_{\psi_i}$, which is called the *routing* function. $\theta_i$ and $\psi_i$ are parameters of $w_{\theta_i}$ and $\pi_{\psi_i}$, respectively. After the policy decomposition with PMOE method, we can rewrite the update rule in Eq. 3 as:

$$\begin{aligned} \theta &= \theta + \nabla_\theta \mathbb{E}_{a_t \sim \pi_{\{\theta,\psi\}}(a_t|s_t)}[Q_\phi(s_t, a_t)], \\ \psi &= \psi + \nabla_\psi \mathbb{E}_{a_t \sim \pi_{\{\theta,\psi\}}(a_t|s_t)}[Q_\phi(s_t, a_t)]. \end{aligned} \tag{5}$$

In practice, we usually apply a Gaussian distribution for either a unimodal policy or the low-level policies here in PMOE, making the overall stochastic policy with PMOE to be a GMM. However, sampling from the distributions of primitives usually embeds a sampling process from a categorical distribution $w$, which makes the differential calculation of policy gradients commonly applied in DRL hard to achieve. We provide a theoretically guaranteed solution for approximating the gradients in the sampling process of PMOE and successfully optimising the PMOE policy model within DRL, which will be described in details.

## 3.3 LEARNING THE ROUTING

To optimise the routing function $w$, which involves a sampling process from a categorical distribution, we propose a frequency loss function to approximate the gradients, which is theoretically proved to approximate $w$ as the probability of the corresponding primitive being the optimal one *w.r.t.* the Q-value function.

Specifically, given a state $s_t$, we sample one action $a_t^i$ from each primitive $\pi_{\psi_i}$, to get a total of $K$ actions $\{a_t^i; i = 1, 2, \cdots, K\}$, and compute $K$ Q-value estimations $\{Q_\phi(s_t, a_t^i); i = 1, 2, \cdots, K\}$ for each of the actions. Then we select an "optimal" primitive index as $j = \arg\max_i Q_\phi(s_t, a_t^i)$. Then we encode a one-hot code vector $v = [v_1, v_2, \cdots, v_K]$ with:

$$v_j = \begin{cases} 1, & \text{if } j = \arg\max_i Q_\phi(s_t, a_t^i); \\ 0, & \text{otherwise.} \end{cases} \tag{6}$$

Here we define a frequency loss function as:

$$\mathcal{L}_{freq} = (v - w)(v - w)^T, w = [w_{\theta_1}, w_{\theta_2}, \cdots, w_{\theta_K}]. \tag{7}$$

We use the proposed frequency loss $\mathcal{L}_{freq}$ as a smooth and differentiable function to update the routing function parameters $\theta$, which is guaranteed to approximate $w_{\theta_i}$ as the probability of the $i$-th primitive being the optimal primitive for current state. Detailed proof is provided in Appendix B.

## 3.4 LEARNING THE PRIMITIVE

To update the $\psi_i$ within each primitive, we provide two approaches of optimising the primitives: *back-propagation-all* and *back-propagation-max* manners.

For the *back-propagation-all* approach, we update all the primitive:

$$\mathcal{L}_{pri}^{bpa} = -\sum_{i}^{K} Q_\phi(s_t, a_t^i), \; a_t^i \sim \pi_{\psi_i}(a_t|s_t). \tag{8}$$

For the *back-propagation-max* approach, we use the highest Q-value estimation as the primitive loss:

$$\mathcal{L}_{pri}^{bpm} = -\max_i \{Q_\phi(s_t, a_t^i; i = 1, 2, \cdots, K)\}, \ a_t^i \sim \pi_{\psi_i}(a_t|s_t). \tag{9}$$

With either approach, we have the stochastic policy gradients as following:

$$\begin{aligned}
\nabla_{\psi_i}\mathcal{L}_{pri} &= -\nabla_{\psi_j}\mathbb{E}_{\pi_{\psi_i}}[Q_\phi(s_t, a_t)] \\
&= \mathbb{E}_{\pi_{\psi_i}}[-Q_\phi(s_t, a_t)\nabla_{\psi_j}\log\pi_{\psi_j}(a_t|s_t)]
\end{aligned} \tag{10}$$

Ideally, both approaches are feasible for learning a PMOE model. However, in practice, we find that the *back-propagation-all* approach will tend to learn primitives that are close to each other, while the *back-propagation-max* approach is capable of keeping primitives distinguishable. The phenomenon is demonstrated in our experimental analysis. Therefore, we adopt the *back-propagation-max* approach as the default setting of PMOE model without additional clarification.

### 3.5 LEARNING THE CRITIC

Similar to standard off-policy RL algorithms, our Q-value network is also trained to minimise the Bellman residual:

$$\mathcal{L}_{critic} = \mathbb{E}_{(s_t, a_t) \sim \mathcal{D}} \|Q_\phi(s_t, a_t) - [r_t + \gamma \max_{a_{t+1}} Q_{\bar{\phi}}(s_{t+1}, a_{t+1})]\|_2, \tag{11}$$

where $\bar{\phi}$ is the parameters of the target network.

The learning component can be easily embedded into the popular actor-critic algorithms, such as soft actor-critic (SAC) (Tuomas et al., 2018), one of the state-of-the-art off-policy RL algorithms. In SAC, $Q_\psi(s_t, a_t^j)$ is substituted with $Q_\psi(s_t, a_t^j) + \alpha\mathcal{H}_j$, where $\alpha$ is temperature and $\mathcal{H}_j - \log\pi_{\psi_j}(a_t|s_t)$ is the entropy which are the same as in SAC. The algorithm is summarised in Algorithm 1. When $K = 1$, our algorithm simply reduced to the standard SAC.

## 4 EXPERIMENTS

In our experiments, we will answer the following questions concerning the advantages of PMOE method and the training settings for achieving good performance:

**Q1**: Is our proposed method more efficient and stable than other model-free RL algorithms?

**Q2**: What is the benefit of using GMM-based policy approximation over unimodal Gaussian policy?

**Q3**: How does the number of primitives affect the performance?

To answer **Q1**, we conduct a thorough comparison on a set of challenging continuous control tasks in OpenAI Gym MuJoCo environments (Brockman et al., 2016) with other baselines, including a MOE method with gating operation (Jacobs et al., 1991b), Double Actor-Critic (DAC) (Zhang & Whiteson, 2019) option framework, and the Multiplicative Compositional Policies (MCP) (Peng et al., 2019). Well-known sample efficient algorithms involving Soft Actor-Critic (SAC) (Tuomas et al., 2018) and Proximal Policy Optimisation (PPO) (Schulman et al., 2017) are adopted as basic DRL algorithms in our experiments, where different multimodal policy approximation methods are built on top. This verifies the generality of PMOE for different DRL algorithms. To answer **Q2**, a deeper investigation of PMOE is conducted to find out the additional effects caused by deploying mixture models rather than single model in policy learning. We start with a simple self-defined target-reaching task to show our method can indeed learn various optimal solutions with distinguishable primitives, which are further demonstrated on complicated tasks in MuJoCo. Additionally, the exploration behaviors are also compared between PMOE and other baselines to explain the advantageous learning efficiency of PMOE. To know how the number of primitives affects the performance (**Q3**), we test PMOE with different different values of $K$ on the *HumanoidStandup-v2* environment. We provide intuitive experiences for determining the number of primitives when applying PMOE. We also demonstrate the advantages of proposed PMOE in its exploration behaviours and the distinctiveness of different primitives learned with PMOE against the baselines.

---

**Algorithm 1:** POME Training Algorithm.

---

**Input:** Policy network $\pi_{\{\theta,\psi\}}$ with parameters $\{\theta,\psi\}$. Critic network $Q_\phi$ with parameters $\phi$.
Initialise target policy network and critic network parameters: $\bar{\theta} \leftarrow \theta, \bar{\psi} \leftarrow \psi, \bar{\phi} \leftarrow \phi$.
Initialise an empty replay buffer: $\mathcal{D} \leftarrow \Phi$.
**while** *not converge* **do**
    **for** *each environment step* **do**
        Sample action from policy: $a_t \sim \pi_{\{\theta,\psi\}}(a_t|s_t)$.
        Interact with the environment: $s_{t+1} = p(s_{t+1|a_t,s_t})$.
        Store data in replay buffer: $\mathcal{D} = \mathcal{D} \cup \{s_t, a_t, s_{t+1}, r(s_t, a_t)\}$.
    **for** *each update step* **do**
        Sample from replay buffer: $\{s_t, a_t, s_{t+1}, r_t\} \sim \mathcal{D}$.
        **for** *each policy update step* **do**
            Sample $K$ actions from primitives: $\{a_t^i \sim \pi_{\psi_i}(a_t|s_t); i = 1, 2, \cdots, K\}$.
            Compute $K$ target Q-value estimations $\mathcal{Q} = \{Q_{\bar{\phi}}(s_t, a_t^i); i = 1, 2, \cdots, K\}$,
              compute primitive loss $\mathcal{L}_{pri}$ according to 9.
            Compute mixing coefficients $w(s_t)$ and one-hot code vector $v$ according to Eq. 6,
              then compute freq-loss $\mathcal{L}_{freq}$ according to Eq. 7.
            Update policy network: $\theta = \theta - \nabla_\theta \mathcal{L}_{freq}, \psi = \psi - \nabla_\psi \mathcal{L}_{pri}$.
        **for** *each critic update step* **do**
            Sample action from mixture policy: $a_{t+1} \sim \pi_{\{\theta,\psi\}}(a_{t+1}|s_{t+1})$.
            Compute next time step target Q-value estimation: $Q_{\bar{\phi}}(s_{t+1}, a_{t+1})$.
            Compute Q-value estimation $Q_\phi(s_t, a_t)$ and compute critic loss $\mathcal{L}_{critic}$ according to
              Eq. 11.
            Update critic network: $\phi = \phi - \nabla_\phi \mathcal{L}_{critic}$.
        Update target network: $\bar{\theta} = \tau\theta + (1-\tau)\bar{\theta}, \bar{\psi} = \tau\psi + (1-\tau)\bar{\psi}, \bar{\theta} = \tau\theta + (1-\tau)\bar{\theta}$.
**Output:** $\theta, \psi, \phi$.

---

### 4.1 COMPARATIVE EVALUATION

The evaluation on the average returns with SAC-based and PPO-based algorithms are shown in Fig. 1 and Fig. 2, respectively. In addition, we compute the AUC (the area under the curves) values for those comparisons in Appendix G. Specifically, in Fig. 1, SAC-based algorithms with either unimodal policy or our PMOE for policy approximation are compared against the MCP (Peng et al., 2019) and gating operation methods (Jacobs et al., 1991b) in terms of the average returns across six typical MuJoCo tasks. In Fig. 2, a comparison with similar settings while basing on a different DRL algorithm is conducted to show the generality of PMOE method. Specifically, we compare the proposed PMOE for policy approximation on PPO with DAC and PPO-based MCP methods. From the comparisons, we found that in relatively simple environments such as *Ant-v2* and *Walker2d-v2*, large $K$ might not be necessary. While complicated environments like *Humanoid-v2* and *HumanoidStandup-v2*, larger $K$ is preferred. This is possible because the optimal solutions might be unique in simple environments thus making multimodal representation unnecessary. Nevertheless, our method is still more sample efficient, showing the capability of degradation of models in our method, where the redundant primitives automatically re-fit onto other necessary primitives and diminish the effective number of $K$. PMOE is testified to provide considerable improvement for general DRL algorithms with stochastic policies on a variety of tasks. Training details are provided in Appendix D.

### 4.2 EXPERIMENT ANALYSIS

We provide in-depth analysis of the proposed PMOE to analysis the differences between PMOE method and other methods in RL process (**Q2**), and estimate the effects of the number of primitives (**Q3**). We use a self-defined *target-reaching* environment to analyse our method. In this environment, the agent starts from a fixed initial position, then acts to reach a random target position within a certain range and avoids the obstacles on the path. Details about this environment is provided in Appendix C.

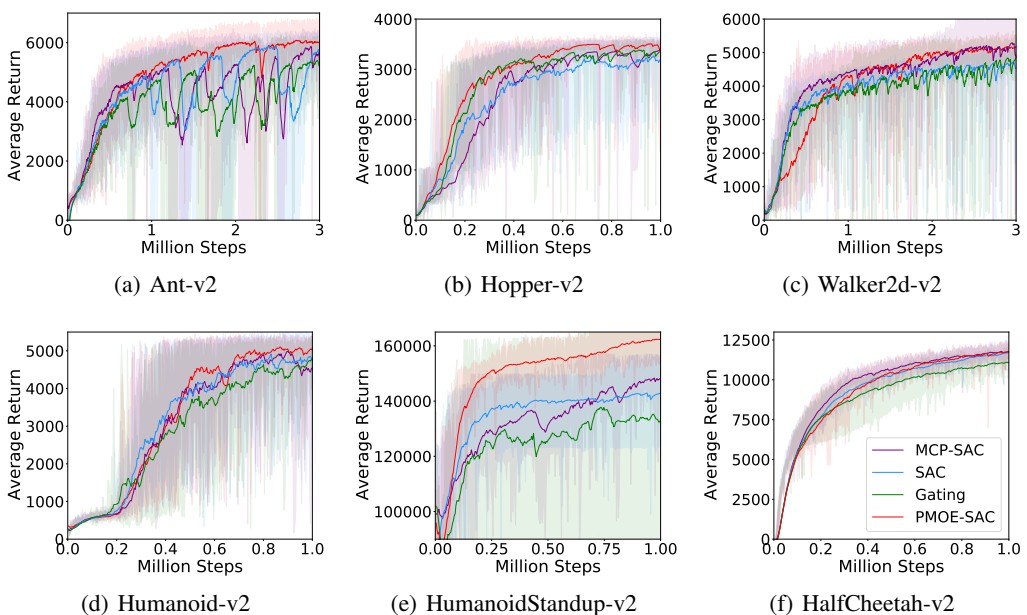

Figure 1: Training curves on MuJoCo benchmarks with SAC-based algorithms. We set PMOE with $K = 4$ in all the experiments except *HalfCheetah-v2* with $K = 2$ and *HumanoidStandup-v2* with $K = 10$.

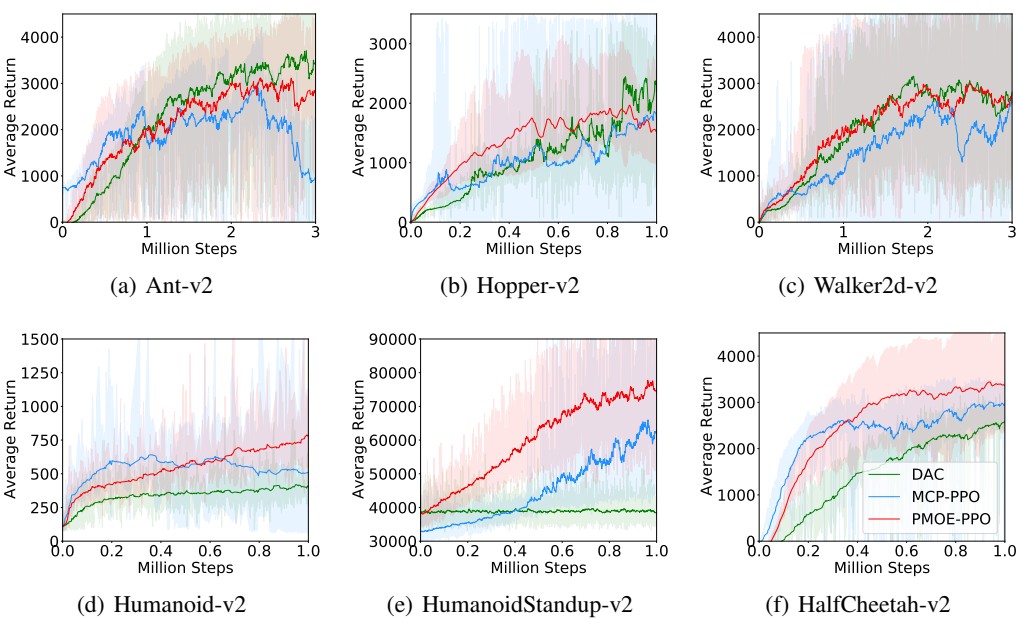

Figure 2: Training curves on MuJoCo benchmark with PPO-based algorithms. We set a larger number, $K = 8$, for the high-dimensional environments *Humanoid-v2* and *HumanoidStandup-v2*, and $K = 4$ for other environments . The results show that our PMOE has comparable performance with the baseline on low-dimension environments, but significantly better than the baseline on high-dimensional environments.

**Distinguishable Primitives.** Fig. 3 demonstrates the distinguishable primitives (Peng et al., 2019) learned with PMOE on the target-reaching environment, for providing a simple and intuitive understanding. After the training stage, we sample 10 trajectories for each method and visualise them in

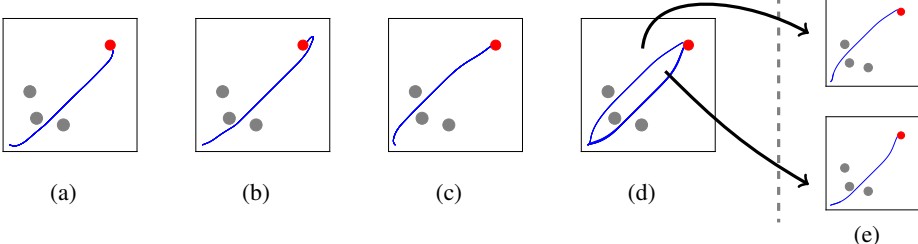

Figure 3: Trajectories of the agents with our method and the baselines in the target-reaching environment. We fix the reset locations of target, obstacles and agent. $(a)$, $(b)$, $(c)$ and $(d)$ visualize the 10 trajectories collected with methods involving: original SAC, gating operation with SAC, back-propagation-all PMOE (discussed in Sec. 3.4) and back-propagation-max PMOE, respectively. $(e)$ shows the trajectories collected with two individual primitives with our approach.

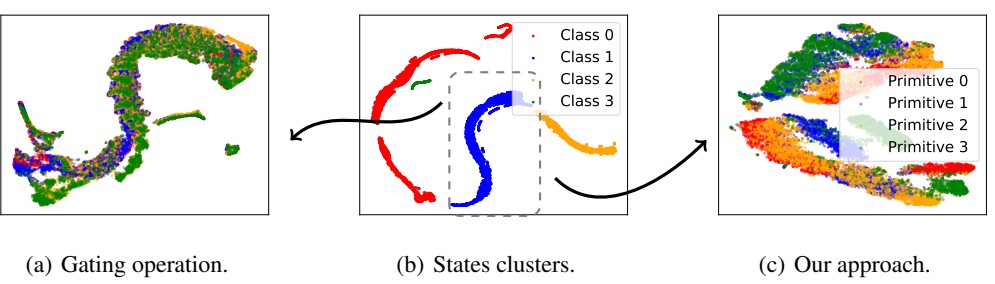

(a) Gating operation.  (b) States clusters.  (c) Our approach.

Figure 4: Visualisation of distinguishable primitives learned with PMOE using t-SNE plot on *Hopper-v2* environment. The states are first clustered as in $(b)$. Then actions within the same state cluster are plotted with t-SNE as in $(a)$ and $(c)$ for the gating method and our approach, respectively. Our method clearly demonstrates more distinguishable primitives for the policy.

Fig. 3. As we can see in Fig. 3(e), PMOE trained in back-propagation-max manner generates two distinguishable trajectories for different primitives.

In Fig. 4, we further demonstrate that PMOE can learn distinguishable primitives on more complex environments with the t-SNE (van der Maaten & Hinton, 2008) method. We sample 10K states $\{s_t; t = 1, 2, \cdots, 10K\}$ from 10 trajectories and use t-SNE to perform dimensionality reduction on states and visualise the results in Fig. 4(b). We randomly choose one state cluster and sample actions corresponding to the states in that cluster. Then we use t-SNE to perform dimensionality reduction on those actions. The reason for taking a state clustering process before the action clustering is to reduce the variances of data in state-action spaces, so that we can better visualise the action primitives for a specific cluster of states. The visualisation of action clustering with our approach and the gating operation are displayed in Fig. 4(a) and Fig. 4(c). More t-SNE visualisations for other MuJoCo environments can be found in Appendix F. Our proposed PMOE method is testified to have stronger capability in distinguishing different primitives during policy learning process. The t-SNE parameters are the same in all the experiments: number of components=2, perplexity=300, learning rate=150, number of iterations=7000.

**Exploration Behaviours.** Fig. 5 demonstrates the exploration trajectories in the target-reaching environment, although all trajectories start from the same initial positions, our methods demonstrate larger exploration ranges compared against other baseline methods, which also yields a higher visiting frequency to the target region (in green) and therefore accelerates the learning process. To some extent, this comparison can explain the improvement of using PMOE as the policy representation over a general unimodal policy. We find that by leveraging the mixture models, the agents gain

effective information more quickly via different exploration behaviours, which cover a larger range of exploration space at the initial stages of learning and therefore ensure a higher ratio of target reaching.

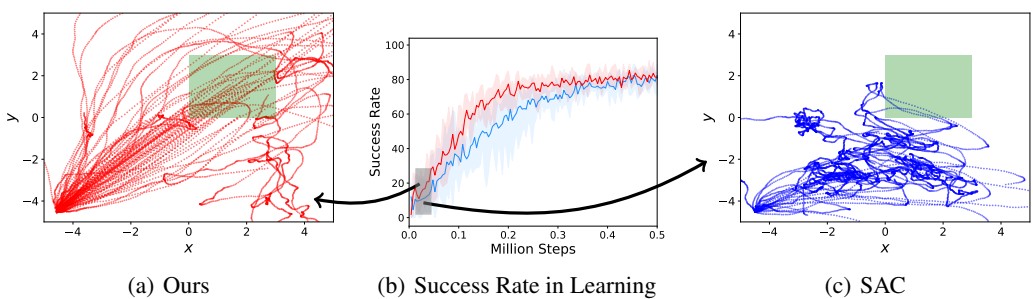

| (a) Ours | (b) Success Rate in Learning | (c) SAC |

Figure 5: Visualisation of exploration trajectories in the initial training stage for the target-reaching environment. The initial $10K$ steps (the grey region on the learning curves in ($b$)) of exploration trajectories are plotted in ($a$) and ($c$) for our PMOE method (red) and SAC (blue), respectively. The green rectangle is the target region.

**Number of Primitives.** We investigate the effects caused by different numbers of primitives in GMM, as shown in Fig. 6. This experiment is conducted on a relatively complex environment — *HumanoidStandup-v2* that has observations with dimension of 376 and actions with dimension of 17, therefore various skills could possibly lead to the goal of the task. The number of primitives $K$ is selected from $\{2, 4, 8, 10, 14, 16\}$. The results show that $K = 10$ seems to perform the best, and $K = 2$ performs the worst among all settings, showing that increasing the number of primitives can improve the learning efficiency in some situations. Our intuitive choice is that the number of primitives needs to be increased when the dimension of action is large.

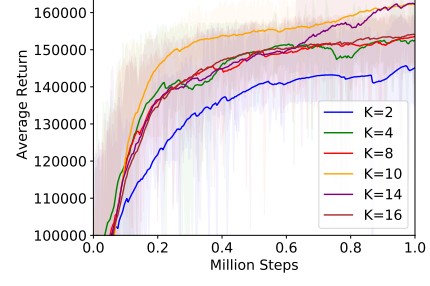

Figure 6: Comparison of different numbers of primitives $K$ in terms of average returns on *HumanoidStandup-v2* environment. For each case, we conduct 5 runs and take the means. The performance increases when $K$ increase from 2 to 10, but decreases if $K$ keep increasing from 10 to 16.

## 5 CONCLUSION

To cope with the problems of low learning efficiency and multimodal solutions in high-dimensional continuous control tasks when applying DRL, this paper proposes the differentiable PMOE method that enables an end-to-end training scheme for generic RL algorithms with stochastic policies. Our proposed method is compatible with policy-gradient-based algorithms, like SAC and PPO. Experiments show performance improvement across various tasks is achieved by applying our PMOE method for policy approximation, as well as displaying distinguishable primitives for multiple solutions.

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

## A    PROBABILISTIC FORMULATION OF PMOE AND GATING OPERATION

In this section, we show a detailed comparison of probabilistic formulation for GMM (as Eq. (12) and (13), used in our PMOE method) and the gating operation method (Eq. (14) to (16)), in term of their PDFs. The gating operation degenerates the multimodal action to a unimodal distribution, which is different from our PMOE method.

For **GMM**, suppose a primitive $\pi_i(s)$ is a Gaussian distribution $\mathcal{N}(a|\mu(s), \sigma^2(s))$, drawing a sample from the mixture model can be seen as the following operation:

$$a \sim \pi(a|s) = \sum_{i=1}^{K} w_i(s)\pi_i(a|s) = \sum_{i=1}^{K} w_i(s)\mathcal{N}(a|\mu_i(s), \sigma_i^2(s)), \tag{12}$$

where the PDF is:

$$p(a) = \sum_{i=1}^{K} \frac{w_i(s)}{\sqrt{2\pi}\sigma_i(s)} \exp\{-\frac{(a - \mu_i(s))^2}{2\sigma_i^2(s)}\}. \tag{13}$$

For **gating operation**, the outputs of the weight operation are the weights of each action from different primitives. With those weights, the gating operation uses the weighted action as the final output action according to Peng et al. (2019):

$$a = \sum_{i=1}^{K} w_i(s)a_i, \ s.t. \ a_i \sim \pi_i(a|s) = \mathcal{N}(a|\mu_i(s), \sigma_i^2(s)). \tag{14}$$

As a primitive is a Gaussian distribution, Eq. 14 becomes:

$$a \sim \mathcal{N}(a| \sum_{i=1}^{K} w_i(s)\mu_i(s), \sum_{i=1}^{K} w_i(s)\sigma_i^2(s)), \tag{15}$$

where the PDF is:

$$p(a) = \frac{1}{\sqrt{2\pi \sum_{i=1}^{K} w_i(s)\sigma_i^2(s)}} \exp\{-\frac{(a - \sum_{i=1}^{K} w_i(s)\mu_i(s))^2}{2 \sum_{i=1}^{K} w_i(s)\sigma_i^2(s)}\}, \tag{16}$$

The above PDF shows that the gating operation could degenerate the Gaussian mixture model into the univariate Gaussian distribution. Other methods (Jacobs et al., 1991a; Peng et al., 2016; Vezhnevets et al., 2017) also have the similar formulation.

## B    PROOF OF FREQUENCY LOSS

In this section, we prove that the proposed frequency loss in our PMOE method produces a consistent estimation of the gradients for optimizing GMMs. The mixing coefficient, also called the weight $w$, stands for the probability of choosing the optimal primitive.

Given the state $s_t$, the frequency of $\arg\max_i Q_\phi(s_t, a_t^i) = j$ being sampled is defined as $f_j$. When we randomly choose $N$ (close to infinity) samples with state $s_t$ from the replay buffer, the number of samples satisfying $\arg\max_i Q_\phi(s_t, a_t^i) = j$ should be $Nf_j$, while the number of those with $\arg\max_i Q_\phi(s_t, a_t^i) \neq j$ is $N(1 - f_j)$. Suppose that we have a loss function for $w_{\theta_j}(s_t)$ as:

$$\begin{aligned} \mathcal{L}_{w_{\theta_j}(s_t)} &= Nf_j[1 - w_{\theta_j}(s_t)]^2 + N(1 - f_j)[0 - w_{\theta_j}(s_t)]^2 \\ &= N[f_j(1 - w_{\theta_j}(s_t))^2 + (1 - f_j)(w_{\theta_j}^2(s_t)] \\ &= N[f_j - 2f_j w_{\theta_j}(s_t) + f_j w_{\theta_j}^2(s_t) + w_{\theta_j}^2(s_t) - f_j w_{\theta_j}^2(s_t)] \\ &= N[w_{\theta_j}^2(s_t) - 2f_j w_{\theta_j}(s_t) + f_j]. \end{aligned} \tag{17}$$

Then the gradient for $\theta_j$ is:

$$\begin{aligned} \nabla_{\theta_j} \mathcal{L}_{w_{\theta_j}(s_t)} &= \nabla_{\theta_j} N[w_{\theta_j}^2(s_t) - 2f_j w_{\theta_j}(s_t) + f_j] \\ &= N[\nabla_{\theta_j} w_{\theta_j}^2(s_t) - 2f_j \nabla_{\theta_j} w_{\theta_j}(s_t)] \\ &= 2N(w_{\theta_j}(s_t) - f_j)\nabla_{\theta_j} w_{\theta_j}(s_t), \end{aligned} \tag{18}$$

Therefore, optimising Eq. 17 is the same as minimising the distance between $w_{\theta_j}(s_t)$ and $f_j$, with the optimal situation as $w_{\theta_j}(s_t) = f_j$ when letting the last formula of Eq. 18 be zero.

## C    Details of Target-Reaching Environment

The visualisation of the target-reaching environment is shown in Fig.7, the blue circle is the agent, the gray circles are obstacles and the circle in red is the target. The agent state is represented by an action vector $a = [a_x, a_y]$ and a velocity vector $v = [v_x, v_y]$. The playground of the environment is continuous and limited in $[-5, 5]$ in both x-axis and y-axis. The agent speed is limited into $[-2, 2]$, the blue coloured agent is placed at position $[x_{ag}, y_{ag}] = [-4.5, -4.5]$ and the red coloured target is randomly placed at position $[x_{tg}, y_{tg}]$, where $x_{tg}, y_{tg} \sim \mathcal{U}(0, 3)$ and $\mathcal{U}$ denotes uniform distribution. There are $M$ gray coloured obstacles with each position $[x_{obs}^i, y_{obs}^i]$, where $x_{obs}^i, y_{obs}^i \sim \mathcal{N}(0, 3^2)$ and $\mathcal{N}$ denotes Gaussian distribution. The observation is composed of $[[x_{tg} - x_{ag}, y_{tg} - y_{ag}], \{[x_{obs}^i - x_{ag}, y_{obs}^i - y_{ag}^i]; i = 1, 2, \cdots, M\}, a, v]$. The input action is the continuous acceleration $a$ which is in the range of $[-2, 2]$.

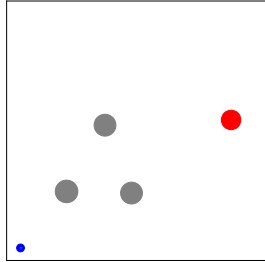

Figure 7: Visualisation of the target-reaching environment

The immediate reward function for each time step is defined as:

$$r = \begin{cases} 100, & \text{if the agent reaches the target;} \\ -10, & \text{if the agent collides with edges or obstacles;} \\ ||v||_2, & \text{otherwise.} \end{cases} \tag{19}$$

## D    Training Details

For PMOE-SAC policy network, we use a two-layer fully-connected (FC) network with 256 hidden units and ReLU activation in each layer. For primitive network $\pi_\psi$, we use a two single-layer FC network, which outputs $\mu$ and $\sigma$ for the Gaussian distribution. Both the output layers for $\mu$ and $\sigma$ have the same number of units, which is $K * dim(\mathcal{A})$, with $K$ as the number of primitives and $dim(\mathcal{A})$ as the dimension of action space, *e.g.*, 17 for *Humanoid-v2*. For the routing function network $w_\theta$, we use a single FC layer with $K$ hidden units and the softmax activation function.. In critic network we use a three-layer FC network with 256, 256 and 1 hidden units in each layer and ReLU activation for the first two layers. Other hyperparameters for training are showed in Tab. 1(a). For PMOE-PPO, we use a two-layer FC network to extract the features of state observations. The FC layers have 64 and 64 hidden units with ReLU activation. The policy network has a single layer with the Tanh activation function. The routing function network has a single FC layer with $K$ units and the softmax activation function. The critic contains one layer only. Other training hyperparameters are showed in Tab. 1(b). We use the same hyperparameters in all the experiments without any fine-tuning. For MCP-SAC, we use the same network structure as MCP-PPO, other training hyperparemeters are the same as shown in Tab. 1(a). For other baselines, we use original hyperparameters mentioned in their paper. The full algorithm is summarised in Algorithm 1.

(a) Hyperparameters for PMOE-SAC

| Parameter | Value |
| --- | --- |
| optimiser | Adam (Kingma & Ba, 2015) |
| learning rate | $10^{-3}$ |
| discount ($\gamma$) | 0.99 |
| replay buffer size | $10^6$ |
| alpha | 0.2 |
| batch size | 100 |
| polyak ($\tau$) | 0.995 |
| episode length | $10^3$ |
| target update interval | 1 |

(b) Hyperparameters for PMOE-PPO

| Parameter | Value |
| --- | --- |
| optimiser | Adam (Kingma & Ba, 2015) |
| learning rate | $3 * 10^{-4}$ |
| discount ($\gamma$) | 0.99 |
| alpha | 0.2 |
| batch size | 64 |
| polyak ($\tau$) | 0.95 |
| episode length | $2 * 10^3$ |
| gradient clip | 0.2 |
| optimisation epochs | 20 |

Table 1: Hyperparameters

## E    PROBABILITY VISUALISATION

We visualise the probabilities of each primitive over the time steps in the MuJoCo *HalfCheetah-v2* environment. As shown in Fig. 8, we found that the probabilities are changed periodically. We also visualise the actions at the selected 5 time steps in one period. As shown in Fig. 9, the primitives are distinguishable enough to develop distinct specialisations.

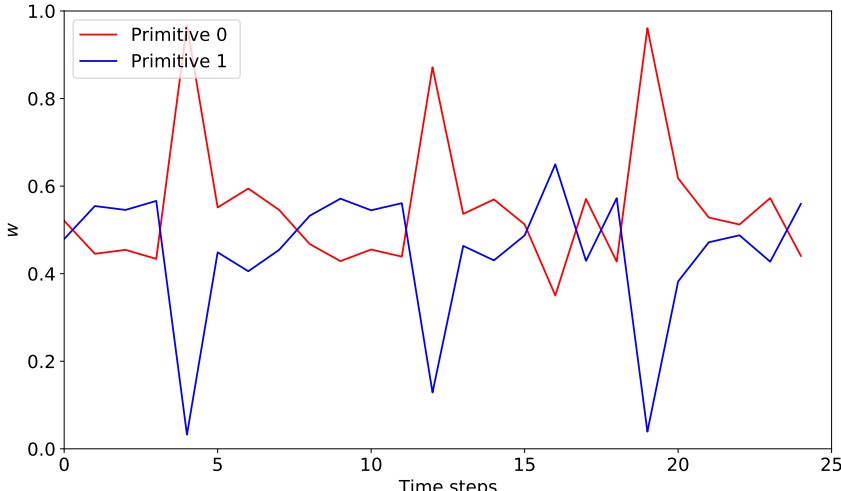

Figure 8: Visualisation of the probabilities of each primitive over the time steps in the MuJoCo *HalfCheetah-v2* environment. The y-axis shows the probabilities of different primitives.

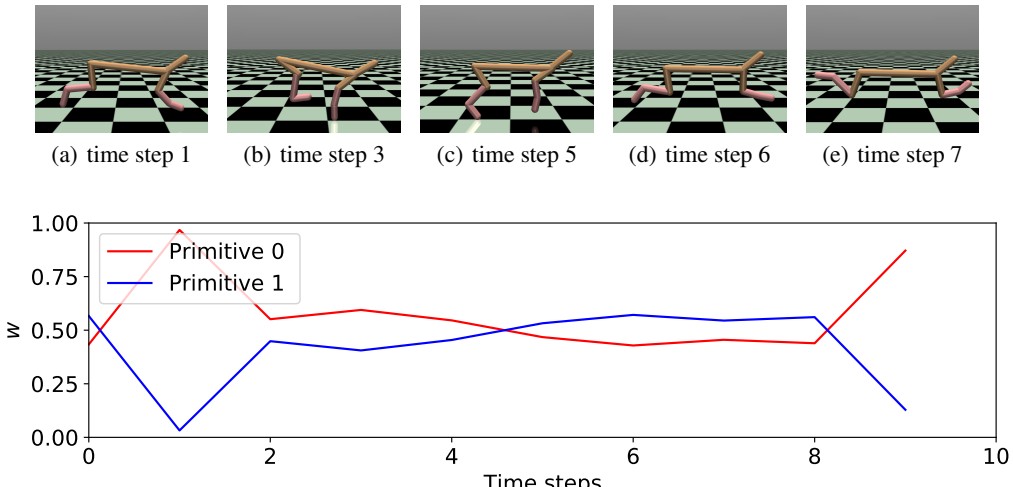

(a) time step 1    (b) time step 3    (c) time step 5    (d) time step 6    (e) time step 7

Figure 9: Visualisation of the actions at the selected 5 time steps in one period. The y-axis shows the probabilities of different primitives. This result shows that the primitives develop distinct specialisations, with the primitive 0 becomes the most active when the front leg touches the ground, while the primitive 1 becomes the most active when the leg leaves the ground.

## F    T-SNE VISUALISATION

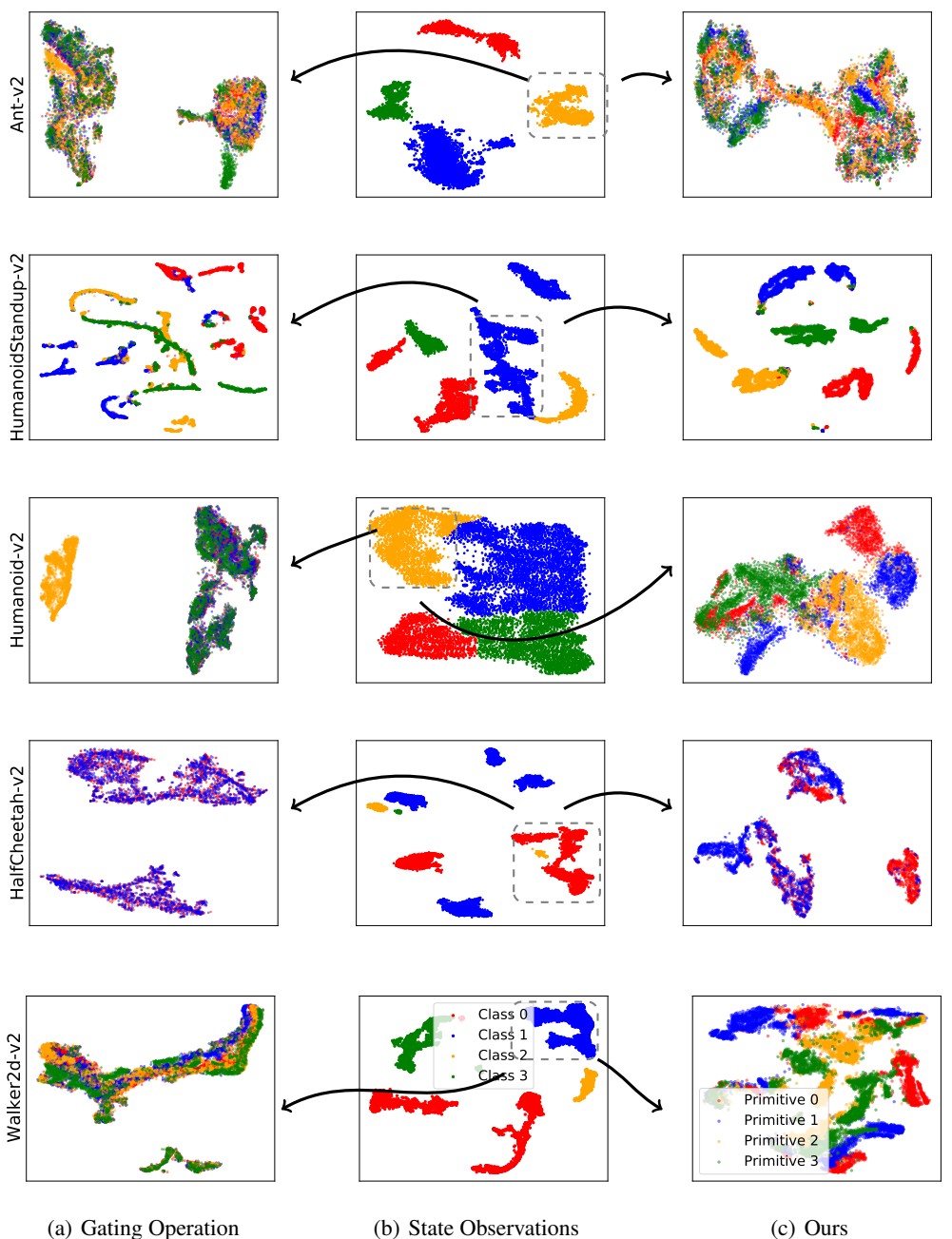

(a) Gating Operation          (b) State Observations          (c) Ours

Figure 10: We plot the t-SNE visualisation for other 5 MuJoCo environments: *Ant-v2*, *HumanoidStandup-v2*, *Humanoid-v2*, *HalfCheetah-v2* and *Walker2d-v2*. Parameters and other details are the same as the setting mentioned in Sec 4.2.

## G    COMPARISON OF AUC

We compute the AUC (the area under the learning curve) to make the figure more readable.

For SAC-based experiments, we assume the AUC of SAC is 1, and AUC values for all methods are shown in Table 2.

| Env | Walker | HalfCheetah | Humanoid | HumanoidStandup | Ant | Hopper |
|---|---|---|---|---|---|---|
| **SAC** | 100% | 100% | **100**% | 100% | 100% | 100% |
| **MCP-SAC** | 104.8% | **103.2**% | 91.2% | 98.2% | 96.4% | 100.5% |
| **Gating** | 97.6% | 95.1% | 84.4% | 92.3% | 89.3% | 109.9% |
| **PMOE-SAC (ours)** | **105.5**% | 99.6% | 94.5% | **113.3**% | **113.4**% | **115.4**% |

Table 2: Comperation of the AUC for the SAC-based methods.

For PPO-based experiments, we assume the AUC of DAC is 1, and AUC values for all methods are shown in Table 3.

| Env | Walker | HalfCheetah | Humanoid | HumanoidStandup | Ant | Hopper |
|---|---|---|---|---|---|---|
| **DAC** | 100% | 100% | 100% | 100% | **100**% | 100% |
| **MCP-PPO** | 75.8% | 158.6% | 159.5% | 116.2% | 83.8% | 93.5% |
| **PMOE-SAC(ours)** | **101.1**% | **171.3**% | **162.2**% | **154.3**% | 93.3% | **123.0**% |

Table 3: Comparison of the AUC for the PPO-based methods.

For the number of $K$ experiments, we assume the AUC for PMOE-SAC with $K = 2$ is 1, so all the methods AUC values are shown in Table 4.

| Number of K | HumanoidStandup |
|---|---|
| **2** | 100% |
| **4** | 107.1% |
| **8** | 106.5% |
| **10** | **111.7**% |
| **14** | 108.0% |
| **16** | 107.0% |

Table 4: Comparison of the AUC as a function of $K$ for PMOE-SAC algorithm on *HumanoidStandup-v2* environment.

## H    COMPARISON WITH OTHER OPTIMIZATION METHODS

As shown in Fig. 11, we compare our methods with the Gumbel-softmax and REINFORCE in one low dimensional MuJoCo task *Hopper-v2* and one high dimensional MuJoCo task *HumanoidStandup-v2*. For all the methods, we randomly choose 5 seeds to plot the learning curves.

## I    COMPARISON WITH PPO

As shown in Fig. 12, we compared with the PPO in the MuJoCo task *HalfCheetah-v2*. For all the methods, we randomly choose 5 seeds to plot the learning curves.

## J    ROBUSTNESS EVALUATION

To evaluate the robustness of our approach, we develop an experiment on the *Hopper-v2* environment. We add a random noise $\epsilon \sim \mathcal{N}(0, \sigma^2)$ to the state observation $s$, and use the noised state observation $\hat{s} = s + \epsilon$ as the input of the policy. Our approach has a better performance in the noised input observation situation, which is shown in Table 5.

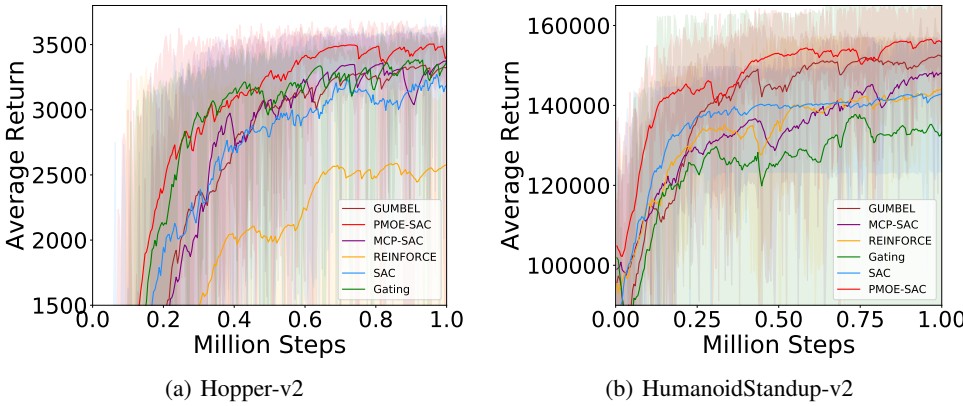

(a) Hopper-v2

(b) HumanoidStandup-v2

Figure 11: Comparison with Gumbel-softmax and REINFORCE in the MuJoCo tasks *Hopper-v2* and *HumanoidStandup-v2*. We found our method PMOE-SAC has a better performance.

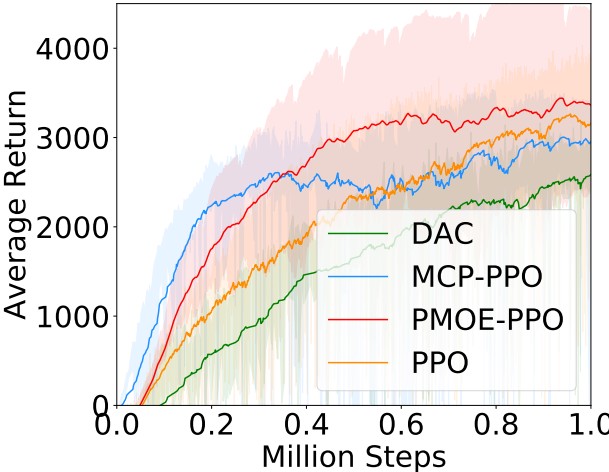

Figure 12: Comparison with PPO in the MuJoCo task *HalfCheetah-v2*. We found our method PMOE-PPO has a better performance.

## K    COMPARISON WITH DIFFERENT PRIMITIVE NUMBERS AND DIFFERENT ENTROPY REGULARISATION

To analyse the relationship of different $K$ and different entropy regularisation(alpha), we compared 7 settings in the MuJoCo task *HumanoidStandup-v2*, where $K$ is the number of primitives and alpha is the entropy regularisation. For each setting, we randomly choose 5 seeds to plot the learning curves in Fig. 13.

| Method | 0 | 0.05 | 0.1 |
|--------|---|------|-----|
| SAC | $3387.4 \pm 2.0$ | $1994.9 \pm 718.6$ | $1006.2 \pm 389.6$ |
| gating | $3444.8 \pm 3.1$ | $2606.9 \pm 864.4$ | $1626.0 \pm 771.9$ |
| MCP | $3524.8 \pm 114.6$ | $1610.2 \pm 357.0$ | $1008.4 \pm 333.4$ |
| Ours | $\mathbf{3632.2 \pm 4.0}$ | $\mathbf{3460.4 \pm 456.7}$ | $\mathbf{1730.0 \pm 703.1}$ |

Table 5: We test our approach in the *Hopper-v2* environment, each column stands for the average return with different variance of the noise distribution, the average return of each methods is averaged over 100 rounds.

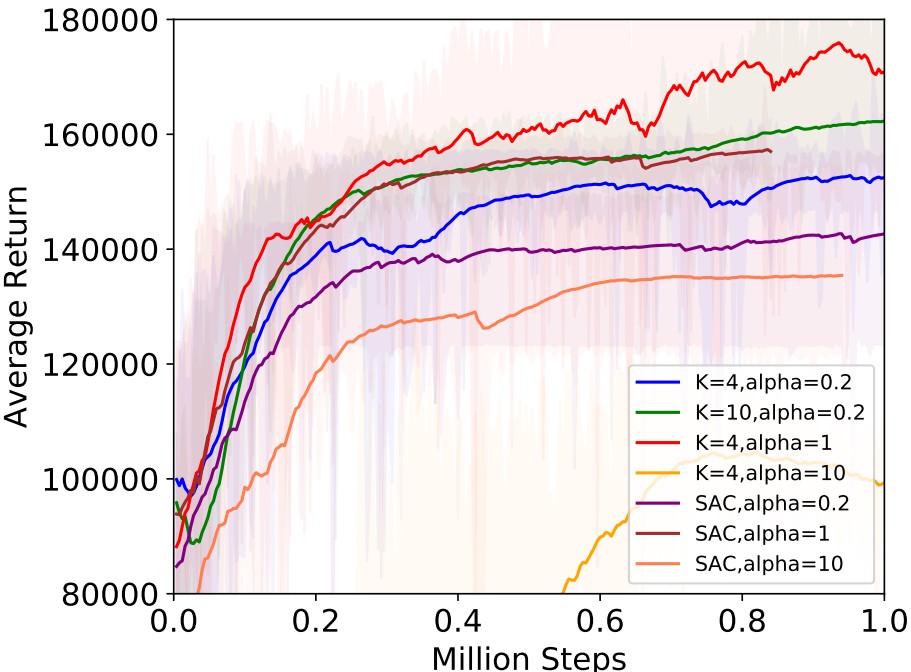

Figure 13: Comparison with different $K$ and different amounts of entropy regularisation. Our approach can be considered as a kind of entropy regularisation method and the number of primitives is positively correlated with the entropy of the policy. The larger number of primitives with smaller entropy has a similar performance to the smaller number of primitives with larger entropy.

