# OpenReview forum: "Probabilistic Mixture-of-Experts for Efficient Deep Reinforcement Learning"
_ICLR.cc/2021/Conference — Reject_

### Official Review · AnonReviewer2 · 2020-10-25
**Paper needs to be more formal**

**Rating:** 4
**Confidence:** 4

**Review:**

The paper studies the problem of differentiating through the policy return when the policy is a Gaussian mixture model. The main contribution of the paper is a heuristic approach for computing this gradient. Having defined the policy update, the authors integrate it to two RL algorithms: PPO and SAC. The experiments show that the algorithms with GMMs behave roughly the same as with a single Gaussian save for one (SAC) or two (PPO) environments. On the other side the authors show that their algorithm can learn multiple solutions on a reaching task and that exploration is better behaved on this task too.

The main problem I had with the paper is the lack of formalism in tackling such a fundamental problem. Given a parameterized GMM model, the contribution of the paper is to study the gradient w.r.t. the GMM’s parameters of an expectation, E[f(X)], where X is sampled from this GMM distribution. It seems like a fundamental problem and authors should discuss more thoroughly existing approaches to computing this gradient. For instance one could simply use the score ratio trick (i.e. REINFORCE) or use importance sampling as in PPO or perhaps a reparameterization trick (with Gumbel-softmax?) if applicable.  At least the two first one are perfectly valid alternatives and should be discussed and ideally compared to the current solution.

In addition it is hard to understand if the proposed solution is an approximation or a heuristic with no guarantees. The authors propose a proof in the appendix but there is no formal statement of what the proof is supposed to show. Is it that the solution is a consistent estimator of the gradient?

In Appendix A, I do not understand what the authors are saying. Somehow they go from a GMM (Eq. 12) to a MoE (Eq. 14-16), i.e. from summing densities to summing random variables. Since these two are very different, what ends up being used in the paper?

I am also confused by the role of the GMM compared to a single Gaussian policy. On one side, the GMM appears to find multiple solutions to an RL problem (Fig. 3), on the other side  the GMM appears to decompose a single solution into sub-policies that are active in different regions of the state-space (Fig. 9). But aren’t these two at odds with each other?

Regarding the exploration advantage, is it known why a GMM should initially better explore than a unimodal Gaussian? Is it task specific or was it also observed on the Mujoco tasks?

Why was there no PPO baseline in Fig. 2? Please also clarify the number of seeds for experiments in Fig. 1 and 2. and discuss statistical significance, since the plots are all very close to each other.

For the Hopper cluster visualization, is the proposed algorithm able to learn different solutions? If that is the case, I think providing a video, or a sequence of images of two different solutions would be more impactful than Fig. 4.

Overall, I think the paper needs significant changes to properly discuss the technical problem it tackles, discuss existing methods, and describe more formally the proposed solution including the theoretical statements provided in App. B. For the experiments, demonstrating that the algorithm is able to simultaneously find multiple solutions is interesting but should be extended to at least one of the highest dimensional locomotion tasks of the paper. For instance showing sampled trajectories for the Hopper as a replacement to Fig. 4 could be very encouraging.

---

> ### Author Response · Authors · 2020-11-25
> **Response to Reviewer 2**
>
> We thank the reviewer for his/her insightful comments and suggestions.
>
> * score ratio trick and reparameterization trick
>
>      We adopt the suggestions from the reviewer and additionally implement the REINFORCE (score ratio trick) and Gumbel-softmax methods for optimizing the GMM models, with comparison results displayed in Appendix H Fig. 11. Importance sampling method as in PPO might not be quite straightforward to implement or less efficient. The results show that PMOE outperforms these methods in thorough evaluation.
>
> * consistent estimator of the gradient
>
>     Yes, our proof in Appendix B shows that the frequency loss function we applied in optimisation will produce consistent estimation of the gradients we expect to have. Additional statements are added in the main paragraph.
>
> * Eq. 12 and Eq. 14-16
>
>     This part is a comparison of mathematical formulas for gating operations and our PMOE method. Detailed statements are added in the modified paper.
>
> * role of the GMM
>
>     We understand the concerns of the reviewer. The distinguishable property of primitives is implicitly provided by the mixture model, so the primitives is indeed not guaranteed to be distinguishable although we observed it in experiments. Actually, it may also happen that the learned MOE can both find multiple solutions and decompose a single solution into different sub-policies or decision regions, considering that the value of K might not always be selected properly. If the potential number of primitives is larger the the possible solutions of a task, the above case can happen. On the contrary, if the number of potential primitives is smaller than the possible solutions, the learned primitives will tend to be distinguishable. Another observation is that, when the potential primitives are more than the solutions, two cases can happen: 1. some primitives will diminish to have nearly zero probability, so that the left primitives will represent distinguishable solutions; 2. some extra primitives will be similar to anchor primitives, so that they may split the decision regions into sub-policies.
>
> * exploration advantage
>
>     Our conjecture is that GMM also has better exploration in MuJoCo tasks. However, we do not have the quantitative results at hand to solidly prove that, which might be left as future work.
>
> * PPO baseline
>
>     PPO baseline is additionally provided in Fig. 12.
>
>
> * videos or sequence of images
>
>     We have add three video clips in the supplementary materials.
>
> * high dimensional locomotion tasks
>
>     Actually, in MuJoCo tasks HumanoidStandup-v2 and Humanoid-v2 are already quite high dimensional tasks, with state dimension of 376 and action dimension of 17. The other MuJoCo tasks are relatively low dimensional tasks, like Hopper, whose state dimension is 11 and action dimension is 3.

---

### Official Review · AnonReviewer1 · 2020-10-28
**Interesting empirical findings for the DRL community**

**Rating:** 6
**Confidence:** 4

**Review:**

I would like to thank the authors of "Probabilistic Mixture-of-Experts for Efficient Deep Reinforcement Learning " for their valuable submission.

Summary of the paper
-
The paper proposes an end-to-end method to train probabilistic mixture-of-experts policies in RL agents. They show that the approach can be applied in the context of popular on-policy and off-policy algorithms, and that it compares favourably (performance and sampling efficiency) to using the same algorithms to train the corresponding unimodal policies. Furthermore they perform an empirically analysis of the individual resulting components and of the impact of these on exploration,

Assessment
-

-- The positives --

The proposed approach is sound and seems to work well in practice.
The empirical evaluation is extensive - the fact that the paper evaluates the proposed approach in combination with different baseline algorithms makes the findings more robust. The analysis is overall quite insightful, especially in terms of understanding the diversity of individual components and the impact of backprop-max vs backprop-all.

-- The concerns --

The paper notes that the mixture of experts seems especially beneficial in high dimensional problems (such as continuous control). This seems an important claim, but it is not clearly backed up. It would be useful to make this statement more quantitative by plotting the improvement in performance over the baseline as a function of the number of dimensions.

The paper notes that the mixture of experts might help by improving exploration. It would be therefore interesting to include a parameter study showing the performance of the baseline algorithm for different amounts of entropy regularisation. This would help to compare the proposed approach to a simpler way of tuning exploration, assess whether a mixture of experts delivers further benefits on top of this (e.g. by providing “deeper” exploration), and allow the reader to compare the sensitivity to the parameter K of the proposed approach to the sensitivity of the baseline to the weight of the entropy regularisation.

Suggestions
--

* Figure 6 could be made more readable by plotting a parameter study instead of a bunch of learning curves, e.g. plot AUC as a function of K.
* It would be helpful for the authors to better discuss the relation, similarity and difference between the proposed approach and popular HRL approaches, in order to better assess the novelty of the method, and to ensure that it is placed in the appropriate context.

Finally, the paper could use one more pass general pass to ensure the writing is fully correct and make it as readable as possible. Please also fix the following typos:
- or without explicit probabilistic representation → the sentence doesn’t connect to the previous one
- Is our method outperform? → does our method outperform?

---

> ### Author Response · Authors · 2020-11-25
> **Response to Reviewer 1**
>
> Thanks for the comments and suggestions by the reviewer.
>
> * benefits not clearly backup up
>
>     It is observed in the MuJoCo tasks, as shown in the Fig.1 and Fig.2, at least for high dimensional environments like HumanoidStandup-v2 (state dimension is 376 and action dimension is 17), our methods outperform all other baselines, while have a comparable performance on low dimensional environments like HalfCheetah-v2 (state dimension is 17 and action dimension is 6).
>
> * different amounts of entropy regularisation
>
>     Thanks for the suggestion. We originally also planned to have a similar analysis. The comparison of our PMOE method with different K values and SAC algorithm with different amounts of entropy are additionally compared in Appendix K.
>
> * AUC as a function of K
>
>     We adopt the suggestions and add the tables of AUC comparisons in Table 2 and Table 3 (both in Appendix G). A table showing AUC with different K values is also added in Table 4 (Appendix G).
>
> * typos
>
>     Modified in the updated paper.

---

### Official Review · AnonReviewer3 · 2020-10-29
**Direct application of mixture-of-experts in RL**

**Rating:** 3
**Confidence:** 5

**Review:**

### Summary
The paper focuses on the policy architecture of deep reinforcement learning algorithms. Specifically, the authors apply the probabilistic mixture-of-experts (PMOE) model in the policy of a reinforcement learning agent, where each primitive is a unimodal Gaussian distribution and the gating model is a simple state-conditioned categorical distribution. The authors derive the corresponding policy gradient objective for the PMOE policy.

The authors apply the PMOE policy on top of SAC and PPO, and perform experiments on the continuous locomotion tasks in the MuJoCo environments. The results indicate that in some tasks, the PMOE policy outperforms the naive policy baseline. The paper also includes ablation studies and visualizations to demonstrate the diversity of the learned primitives of the PMOE policy.


### Comments
The paper is well written and the idea proposed in this paper is really easy to follow. The authors also include a wide suite of experiments with both on-policy and off-policy RL algorithms to demonstrate the performance of the proposed method, and various ablation studies and visualizations to demonstrate the behavior of PMOE policy. Despite these advantages, I cannot recommend acceptance of this paper due to the lack of novelty and significant performance improvement.

First of all, as the experiment results in this paper suggest, the performance gain of the PMOE policy is marginal and highly task-specific. Only in one of the 6 tasks the PMOE policy exhibits significant benefit over baselines. Therefore, from the scope of experiments in this paper, it is hard to conclude that PMOE policy really has meaningful advantages over a naive policy parameterization.

Moreover, as described in the paper, the PMOE model is a fairly well-studied model, and it seems like the only contribution in this paper is the application of it in the reinforcement learning setting. I’m not convinced that such a straightforward application has enough contribution, especially given the fact that the performance improvement is not significant.

Therefore, due to the lack of novelty and significant performance improvement, I cannot recommend acceptance of this paper.

---

> ### Author Response · Authors · 2020-11-25
> **Response to Reviewer 3**
>
> We thank the reviewer for his/her insightful comments and suggestions.
>
> * no significant performance improvement
>
> Firstly, our method is not worse than other baselines in almost all the tasks, and in some tasks, our method has better performance, especially in the high dimensional MuJoCo environments like HumanoidStandup-v2. Besides, our method has better exploration efficiency as showed in Fig. 5 and distinguishable property as showed in Fig. 3-4. We also added a robustness evaluation in the Appendix J, showing that our approach has a better performance in situations with noised input observation.
>
> * lack of novelty
>
> As discussed with Reviewer 2, to tackle with the indifferentiable problem in optimizing the GMM, we propose the frequency loss to compare against several existing approaches (REINFORCE, Gumbel-softmax) in our experiments. We also testify that our PMOE method outperforms other MOE methods and hierarchical policy methods, and these methods face problems like model degeneration and low-efficiency. Thus, our proposed method approximates the mixture model without degeneration, we found our method has advantages in exploration efficiency, distinguishable property, potential in the high dimensional environment as showed in experiments. In addition, we proposed a frequency loss to guarantee the probability of each primitive to output the best action for the current state, with a consistent estimation of gradients, which is proved in Appendix B.

---

### Official Review · AnonReviewer4 · 2020-10-30
**Solid work however an extension**

**Rating:** 6
**Confidence:** 4

**Review:**

In this paper, the authors propose to use mixture models of policies and present good experimental results. The approach is quite sound, and the experimental results looks solid.

However, there are some concerns that make this paper cannot be fully appreciated:
1. The first and second listed contributions are about "the undifferentiability problem", and the third is a verification of the propose algorithm.
    However, a more fundamental question would be the following ones:
    A). Why the author would like to propose mixture models and hope the community to adopt this framework, to further enhance the DRL performance, right?    I agree the mixture models have great potential, but could you please justify your topic/target issue first, and then present your solution?   I would say, if the authors naturally expect the reviewer or readers to naturally believe in what you did, this is not right. Please show your intellectual contributions explicitly, and use them to convince readers.
         Now, from a reader's perspective, mixture model is well-established, and this work extends it a little.  I agree mixture models are promising, but I am not convinced by this work.  How would the authors respond?
         Since, it is some kind of extension, the novelty cannot be fully appreciated.
    B). In terms of algorithm, is there some innovative contribution?  It is hard to tell, I think the authors would also agree that such steps are straightforward.
         Therefore, will the experiments solely enough to justify an acceptance?
    C). Regarding your experiments, besides the three questions raised by the authors (which are reasonable).
          I would ask some questions beyond, would be mixture model be more powerful or practically important to address the robustness, stability issues that faced by current DRL algorithms?  (Q1 mentioned stability, while the figures do not fully address it, right?)
          Also, as for mixture models, would an adaptive algorithms be more appropriate solution?   Since mixture model allows more freedom to balance exploration and exploitation, and an adaptive scheme would allocate exploration budget to more informative policies or state regions?
          Therefore, the current experiments do not fully justify the value of this paper.  I have concerns.

---

> ### Author Response · Authors · 2020-11-25
> **Response to Reviewer 4**
>
> Thanks for the comments and suggestions by the reviewer.
>
> * experiments
>
> We conduct thorough experiments in PMOE method for both on-policy and off-policy RL algorithms, over from low-dimensional to high-dimensional tasks with continuous control. We further testify that PMOE for policy function approximation has  advantages in several aspects: good learning efficiency as showed in Fig. 1-2, distinguishable property as showed in Fig. 3-4, better exploration ability as showed in Fig. 5. Some additional experiments are added in Appendix G to J.
>
> * robustness and stability
>
> We added an robustness evaluation in the Appendix J, to show that our approach has a better performance in situations with noised input observations.
>
> * Why not use an adaptive algorithms
>
> We considered this approach, however, to implement the adaptive scheme, we need to make the primitives more distinguishable and have a proper metric to measure the similarity between primitives, which is beyond the discussion of this paper. But we will investigate how to implement the adaptive scheme in the future work.

---

### Decision · Program_Chairs · 2021-01-07
**Final Decision**

**Decision:**

Reject

**Comment:**

The paper studies mixture of expert policies for reinforcement learning agents, focusing on the problem of policy gradient estimation. The paper proposes a new way to compute the gradient, apply it to two reinforcement learning algorithms, PPO and SAC, and demonstrate it in continuous MuJoCo environments, showing results that are comparable to or slightly exceeds unimodal policies. The main issue raised by multiple reviewers is novelty. Mixture of expert models have been widely studied in the context of reinforcement learning, and while the paper proposes a new method for the gradient computation, a more suitable format, as pointed out by Reviewer 2, could be to ground the paper around the proposed gradient estimator, and compare, both analytically and empirically, it to existing alternatives. Therefore, I recommend rejecting this submission.